# Geochemical and Sr-Isotopic Study of Clinopyroxenes from Somma-Vesuvius Lavas: Inferences for Magmatic Processes and Eruptive Behavior

**Valeria Di Renzo [1,\*], Carlo Pelullo [2], Ilenia Arienzo [2], Lucia Civetta [1], Paola Petrosino [1] and Massimo D'Antonio [1,\*]**

[1] Dipartimento di Scienze della Terra, dell'Ambiente e delle Risorse, Università degli Studi di Napoli Federico II, Via Vicinale Cupa Cintia, 21, 80126 Napoli, Italy

[2] Istituto Nazionale di Geofisica e Vulcanologia, Sezione di Napoli Osservatorio Vesuviano, Via Diocleziano, 328, 80124 Napoli, Italy

[\*] Correspondence: valeria.direnzo@unina.it (V.D.R.); massimo.dantonio@unina.it (M.D.)

**Abstract:** Somma-Vesuvius is one of the most dangerous active Italian volcanoes, due to the explosive character of its activity and because it is surrounded by an intensely urbanized area. For mitigating the volcanic risks, it is important to define how the Somma-Vesuvius magmatic system worked during the past activity and what processes took place. A continuous coring borehole drilled at Camaldoli della Torre, along the southern slopes of Somma-Vesuvius, allowed reconstructing its volcanic and magmatic history in a previous study. In this work, the wide range of chemical (Mg# = 92–69) and isotopic ($^{87}Sr/^{86}Sr$ = 0.70781–0.70681) compositions, collected on single clinopyroxene crystals separated from selected lava flow units of the Camaldoli della Torre sequence, have been integrated with the already available bulk geochemical and Sr-isotopic data. The detected chemical and isotopic signatures and their variation through time allow us to better constrain the behavior of the volcano magmatic feeding system, highlighting that mixing and/or assimilation processes occurred before a significant change in the eruptive dynamics at Somma-Vesuvius during a period of polycyclic caldera formation, starting with the Pomici di Base Plinian eruption (ca. 22 ka).

**Keywords:** Somma-Vesuvius; Camaldoli della Torre deep borehole; Sr-isotopes; clinopyroxene; open-system magmatic processes; eruptive behavior





## 1. Introduction

Somma-Vesuvius (Figure 1) is one of the best studied active volcanoes in the world because of the high risk related to the intense urbanization at its footslopes. It is a volcanic complex formed by an older strato-volcano, Mt. Somma, cut by an eccentric, polyphasic caldera, and by a younger strato-cone, Mt. Vesuvius, grown inside the caldera during historical times (e.g., [1,2] and references therein). The last eruption, which occurred in 1944 AD, was both effusive and explosive, threatened the inhabitants of the San Sebastiano al Vesuvio town with a lava flow and covered a large part of the southern Italian peninsula with a thin ash deposit [3].

In the past few decades, the magmatic system feeding Somma-Vesuvius over time has been the subject of several volcanological, geochronological, petrological and geochemical scientific articles (e.g., [1,5–24]). The most recent and exhaustive investigation has been eased by a borehole drilled in the southern slope of Somma-Vesuvius, at Camaldoli della Torre (CdT; Figure 1), that allowed reconstructing the ancient history of the Somma-Vesuvius magmatic system, through bulk rock geochemical (major oxides and trace elements) and Sr-Nd-Pb-B isotopic data [16]. In this work, we present a detailed geochemical investigation of the magmatic processes that occurred between 40 ka and 19 ka

ago, a timespan that includes the Pomici di Base eruption (ca. 22 ka; [7]; [14]C age recalibrated by Santacroce et al. [20]), the first Plinian event attributed to the Somma-Vesuvius. The volcanic products of the studied time period are characterized by geochemical and isotopic variations suggesting an open-system evolution of the magmas that deserves to be understood thoroughly. To this end, we employed microanalytical isotopic techniques to acquire new $^{87}$Sr/$^{86}$Sr data on single clinopyroxene crystals separated from samples of the CdT cored sequence of different ages and compositions (Figure 2) described by Di Renzo et al. [16].

The rationale of this research is that clinopyroxene is a ubiquitous mineral phase in the Somma-Vesuvius products and should retain the isotopic composition of the magma from which it was segregated. A simple way to characterize the Sr-isotopic composition of clinopyroxene is by collecting several crystals from a crushed fraction of a rock by hand-picking under a microscope, leaching and analyzing them as a "bulk" sample. Bulk clinopyroxene crystals separated from a few volcanic units of the CdT sequence have already been analyzed for Sr-isotopes providing preliminary evidence of isotopic disequilibrium [16]. However, isotopic analyses carried out on bulk mineral separates could likely mask the variability of different crystal populations grown in isotopically variable magmas, in the event of more complex magmatic processes e.g., [25–28]. Therefore, analysis of single crystals, namely a few individual clinopyroxene crystals separated from a rock, each of them analyzed as a single sample, should allow better investigating the open-system magmatic processes that occurred in the Somma-Vesuvius plumbing system and identifying possible mixing endmembers. The investigation of such processes is useful since the knowledge of the behavior of the magmatic system is crucial for both interpreting any possible change in the dynamics of the volcano and predicting its future behavior.

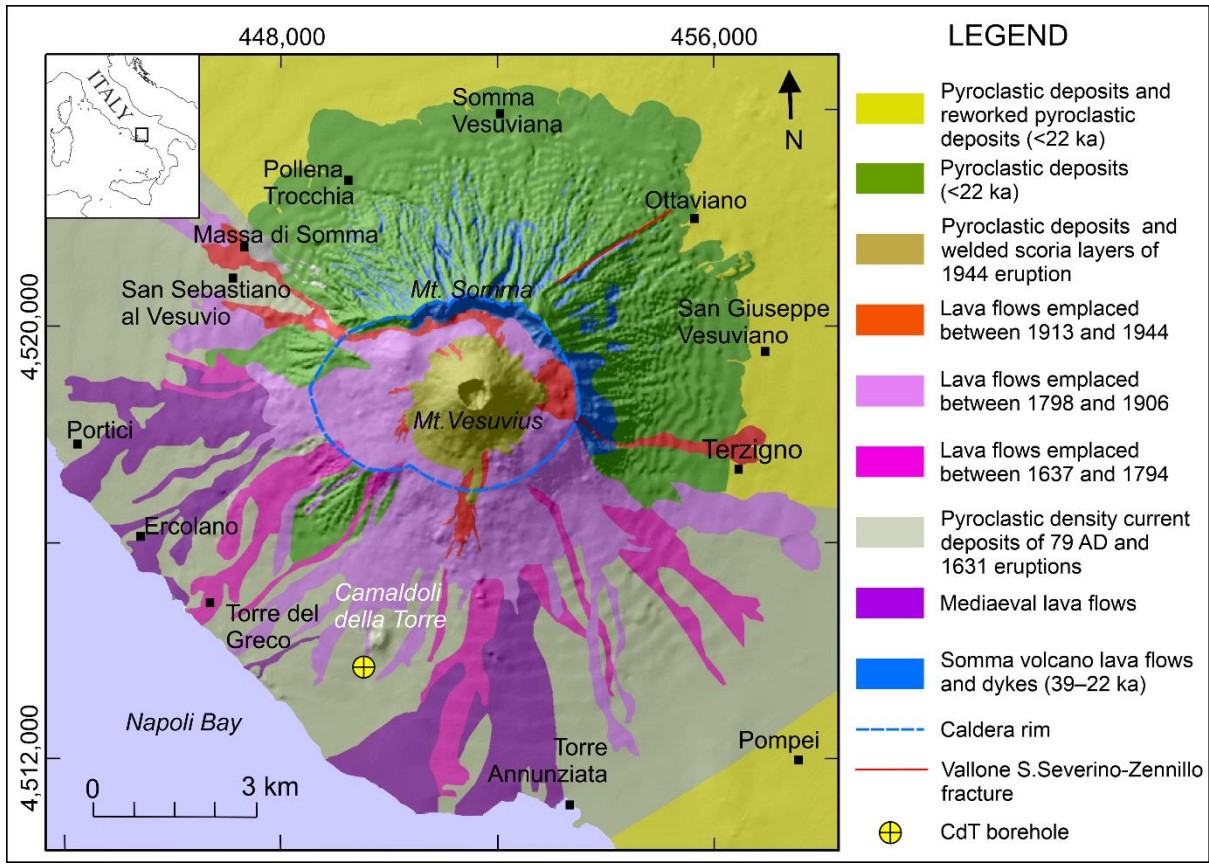

**Figure 1.** Geological map of Vesuvius with the location of the Camaldoli della Torre borehole (modified after [4]).

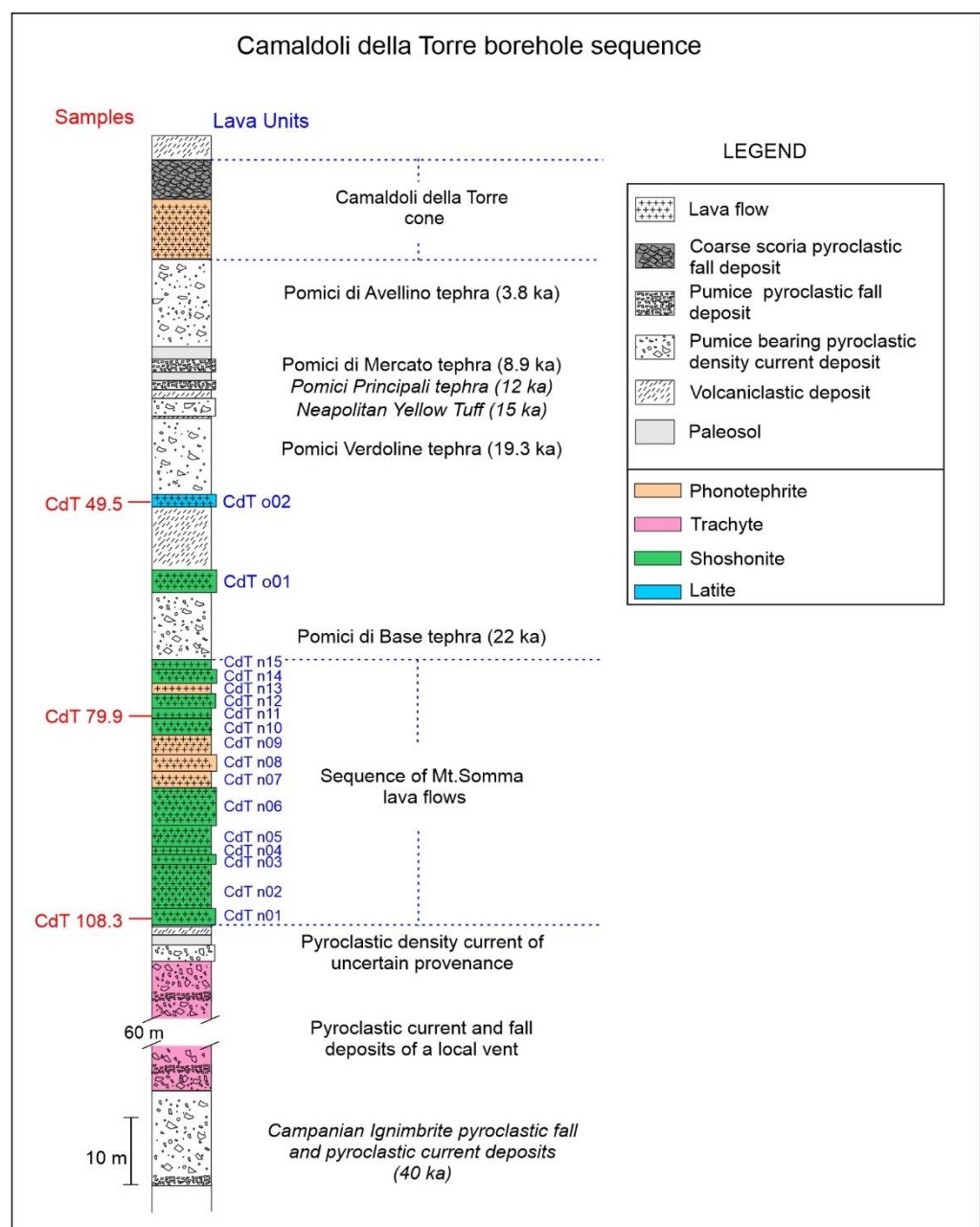

**Figure 2.** Portion of the CdT borehole stratigraphic sequence, showing the position of the investigated volcanic units (redrawn after [16]). Products from Campi Flegrei eruptions are shown in italics.

## 2. Volcanological and Petrological Background

The volcanic activity in the Vesuvian area began ca. 400 ka ago [29], although the present volcano formed after the huge Campanian Ignimbrite (CI) eruption [30], which occurred at Campi Flegrei ca. 40 ka ago (e.g., [31] and references therein). Since 40 ka (Figure 2), the volcano has been characterized either by long periods of quiescence, interrupted by Plinian or sub-Plinian eruptions, or by periods of semi-persistent mildly explosive activity. The latter were frequently interrupted by violent explosive–effusive eruptions, related to the alternation of closed and open conduit conditions, respectively, building the strato-volcano (e.g., [6]). The last period of persistent volcanism started after the 1631 AD sub-Plinian eruption and lasted until 1944 AD. Since that time, Vesuvius has been quiescent, as it has not shown clear signs of unrest; only moderate seismicity and fumaroles testify to its activity.

At Somma-Vesuvius, three main rock series were generated by volcanic activity and can be distinguished based on different ages, degrees of silica undersaturation and degrees of alkali enrichment [5]. The rocks older than about 19 ka mostly consist of poorly silica undersaturated with minor oversaturated rocks ranging from potassic trachybasalt to trachyte. A second series (from about 19 ka to 79 AD) essentially consists of phonotephrites, tephriphonolites and phonolites with intermediate degrees of silica undersaturation. A third series, younger than 79 AD, shows strongly undersaturated leucite-bearing phonotephrite, tephriphonolite to minor phonolite composition [5,8,11,14–18].

In the cored CdT sequence (Figure 2), widely distributed volcanic units, such as the CI from the Campi Flegrei area and some Somma-Vesuvius Plinian deposits, were drilled. Their study helped constrain the relative age of the various deposits and investigate the chemical–physical variations of the magmatic feeding system [16]. At the base of the sequence, trachytic and phonolitic pyroclastic units were found, some of them clearly produced by local vents, older than the CI. Above the CI products, the sequence includes: the trachytic deposits of a local tuff cone; a sequence of marine sediments and mature paleosols; and a succession of at least 17 lava flows shoshonitic, latitic and tephriphonolitic in composition, related to the Mt. Somma activity, which occurred up to ca. 19 ka (Figure 2). This sequence is covered by the Pomici Verdoline (19,265 $\pm$ 105 cal years BP; [20]), Pomici di Mercato (8890 $\pm$ 90 cal years BP; [7]) and Avellino (3818–3859 cal years BP; [32]) Plinian eruption products (Figure 2). The Avellino products are overlain by tephriphonolitic lava and scoria layers produced by the Camaldoli della Torre spatter cone, which closes the drilled sequence (Figure 2). As inferred from the stratigraphic relations, the age of the CdT cone is hence comprised between ca. 3800 years BP and 79 AD, i.e., the age of the overlying Pompei eruption products.

Di Renzo et al. [16] performed a thorough petrological characterization of the volcanic units of the CdT succession attributed to the Somma-Vesuvius volcano. The authors illustrated that the products are variably K-alkaline and silica-undersaturated and include shoshonites, latites, trachytes, phonolites and phonotephrites. Bulk geochemical data indicate variable degrees of enrichment in incompatible trace elements. Sr-Nd-Pb-B isotope data are variable as well and suggest open-system evolution processes superimposed to dominant fractional crystallization processes. The isotopic variations were interpreted [16] as the result of different processes: source heterogeneity and assimilation of either Hercynian continental crust or Mesozoic limestone during magma ascent to the surface and stagnation at mid-crustal depth. Results of isotopic and trace element data modeling carried out by Di Renzo et al. [16] showed that contamination by Hercynian crust satisfactorily explains the isotopic variation over the shoshonite–latite transition, with 2% assimilation and 18% crystallization. The shoshonite–phonotephrite transition instead was suggested to be the result of contamination by Mesozoic limestone. At shallower depths, the mixing of magmas with variable isotopic imprint possibly determined the wide isotopic range of the Somma-Vesuvius products [16].

## 3. Materials and Methods

The selected samples, labeled CdT 108.3, CdT 79.9 and CdT 49.5 according to the stratigraphic depth, were taken from units named CdTn01, CdTn11 and CdTo02 belonging to the CdT succession (Figure 2; [16]). These units are three lava flows emplaced between the CI and the Pomici Verdoline eruptions (from ca. 40 ka to 19.3 ka). Samples CdT 108.3 (K-trachybasalt/shoshonite) and CdT 79.9 (shoshonite) are part of a sequence of 15 lava flows emplaced during the building phase of the Mt. Somma strato-volcano, between ca. 40 ka and ca. 22 ka, the age of the Pomici di Base eruption [7,20]. This event marks the beginning of a caldera formation phase due to a succession of several major Plinian eruptions (Pomici di Base, Pomici di Mercato, Pomici di Avellino and Pompei). Summit caldera collapses occurred after each Plinian eruption (e.g., [1]). After the Pomici di Base eruption, the activity resumed along several eruptive fractures with the emission of K-latitic products through eccentric vents in both the northern (lava flows and scoriae from Vallone

San Severino-Zennillo at Ottaviano and Alveo di Pollena) and southern (Camaldoli della Torre cinder cone; [5]) sectors of the volcano. Sample CdT 49.5 belongs to a latitic lava underlying the deposits of the Pomici Verdoline sub-Plinian eruption (19.3 ka; [20]).

The selected lava samples were gently crushed to lapilli-size grains through a jaw crusher. Samples were sieved using a stack of sieves with meshes ranging from 0.5 to 4 mm. From the sieved aliquots, clinopyroxene phenocrysts were hand-picked under a binocular microscope. Some of the hand-picked crystals were mounted on epoxy resin and polished (Figure 3) for both petrographic observations and chemical microanalysis by SEM-EDS techniques at DiSTAR, University of Napoli Federico II. The average size of the crystals from sample CdT 108.3 is $1.6 \pm 0.3$ mm, that of clinopyroxene from sample CdT 79.9 is $2.9 \pm 0.6$ mm and that of crystals from sample CdT 49.5 is $1.3 \pm 0.3$ mm. The maximum size of the individuals is 2.8 mm (sample CdT 108.3), 4.2 mm (sample CdT 79.9) and 1.4 mm (sample CdT 49.5). The crystals were carefully observed in reflected and transmitted light under an optical microscope before the analysis. The chemical composition of major (Si, Ti, Al, Fe, Mg and Ca) and minor (Mn, Na and Cr) oxides was determined using a Zeiss Merlin VP Compact scanning electron microscope (SEM) coupled with an Oxford Instruments Microanalysis Unit equipped with an INCA X-act detector for EDS. Measurements were performed at a 15 kV primary beam voltage, 50–100 µA filament current, variable spot size and 10 s net acquisition time. The following Smithsonian Institute and MAC (Micro-Analysis Consultants Ltd., Saint Ives, UK) standards were used for calibration: diopside (Ca), fayalite (Fe), San Carlos olivine (Mg), anorthoclase (Na, Al, Si), rutile (Ti), serandite (Mn), microcline (K), apatite (P), fluorite (F), pyrite (S) and sodium chloride (Cl). Relative analytical uncertainty is typically ~1% for major elements and ~3–5% for minor elements. The results of clinopyroxene analysis are listed in Table S1 of the Supplementary Materials.

Six single crystals separated from the trachybasalt/shoshonite CdT 108.3, eight from the shoshonite CdT 79.9 and seven from the latite CdT 49.5 were selected for Sr-isotopic analysis. Each single crystal to be analyzed was washed in an ultrasonic cleaner with Milli-Q $H_2O$. Sample digestion was achieved by using ultrapure HF, $HNO_3$ and HCl mixtures. After dissolution, samples were loaded on micro-columns filled with 70 µL of Sr-Resin (TrisKem International) 50–100 µm, and their Sr fraction was extracted by using the chromatographic separation scheme proposed by Charlier et al. [33]). An aliquot of each dissolved single clinopyroxene crystal was analyzed by inductively coupled plasma atomic emission spectrometry (ICP-AES) to determine rough Mg and Fe contents, useful for a semi-quantitative evaluation (courtesy of Professor D. Morgan). Both the chemical procedure and the ICP-AES determinations were performed at the Department of Earth Sciences, Durham University, UK.

The Sr-isotopic ratios were determined at the Radiogenic Isotope Laboratory of the Istituto Nazionale di Geofisica e Vulcanologia, sezione di Napoli-Osservatorio Vesuviano. They were acquired in static mode by thermal ionization mass spectrometry techniques using a Thermo Finnigan Triton TI mass spectrometer. Each measurement included 180 acquisitions (cycles) of the relevant isotope ratios over a ca. 4 s integration time each. During the time of isotopic data acquisition, aliquots of NIST-SRM 987 international standard with approximate Sr content of 3, 6 and 12 ng gave a mean value of $^{87}Sr/^{86}Sr = 0.71024 \pm 0.00002$ ($2\sigma$, N = 32) (Table S2 of Supplementary Materials). External reproducibility ($2\sigma$) during the period of measurements was calculated according to Goldstein et al. [34]. During acquisition, Sr-isotopic ratios were normalized for within-run isotopic fractionation to $^{86}Sr/^{88}Sr = 0.1194$. The measured Sr isotopic ratios were normalized to the accepted value of NIST SRM 987 international standard ($^{87}Sr/^{86}Sr = 0.71025$; [35]). Data on weight, approximate Mg and Fe contents (ICP-AES) and Sr isotopic composition of the analyzed clinopyroxene phenocrysts are reported in Table 1.

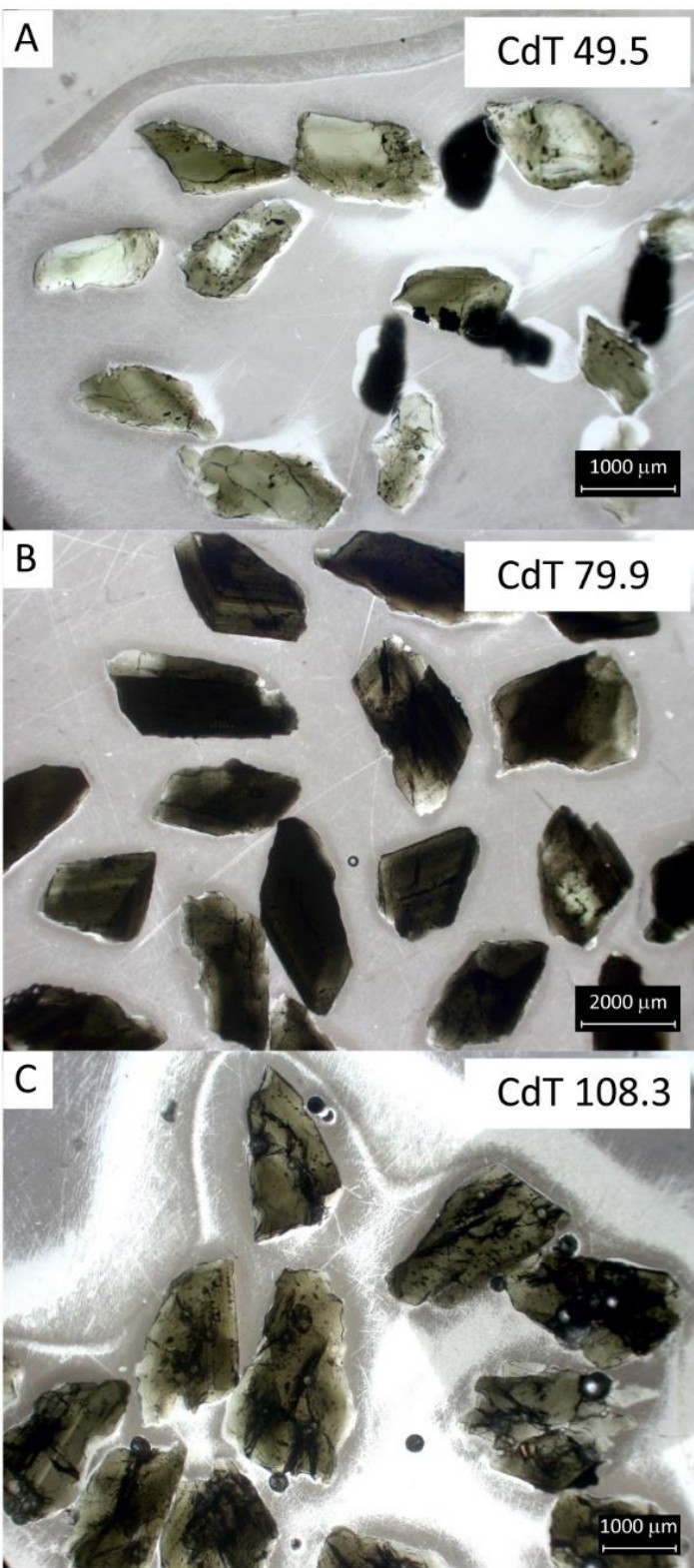

**Figure 3.** Photomicrographs of clinopyroxene phenocrysts from the investigated CdT units, mounted on epoxy resin for petrographic observations and SEM-EDS microanalysis. The panels refer to the clinopyroxene fractions of the three analyzed samples. (**A**) sample Cdt 49.5; (**B**) sample Cdt 79.9; (**C**) Cdt sample 108.3.

**Table 1.** Sr-isotopic ratios of single clinopyroxene crystals of Mt. Somma-Vesuvius lavas selected from the CdT borehole sequence. Mg and Fe contents are approximate values determined by ICP-AES (see text).

| Sample ID | Weight (mg) | Fe (ppm) | Mg (ppm) | Fe Molar | Mg Molar | Mg# * | $^{87}$Sr/$^{86}$Sr ** | 2se *** |
|---|---|---|---|---|---|---|---|---|
| CdT 49.5 | | | | | | | | |
| 49.5 cpx2 | 0.7075 | 0.323 | 0.683 | 0.006 | 0.028 | 82.4 | 0.70752 | 0.00002 |
| 49.5 cpx3 | 0.7294 | 0.475 | 0.563 | 0.009 | 0.023 | 71.9 | 0.70759 | 0.00002 |
| 49.5 cpx4 | 0.7509 | 0.400 | 0.683 | 0.007 | 0.028 | 80.0 | 0.70768 | 0.00002 |
| 49.5 cpx5 | 0.2916 | 0.197 | 0.222 | 0.004 | 0.009 | 69.2 | 0.70763 | 0.00003 |
| 49.5 cpx6 | 0.3205 | 0.274 | 0.269 | 0.005 | 0.011 | 68.8 | 0.70765 | 0.00003 |
| 49.5 cpx7 | 0.9015 | 0.315 | 0.522 | 0.006 | 0.021 | 77.8 | 0.70780 | 0.00003 |
| 49.5 cpx8 | 0.2121 | 0.142 | 0.163 | 0.003 | 0.007 | 70.0 | 0.70781 | 0.00002 |
| CdT 79.9 | | | | | | | | |
| 79.9 cpx1 | 6.2690 | 3.337 | 4.720 | 0.060 | 0.194 | 76.4 | 0.70712 | 0.00001 |
| 79.9 cpx2 | 4.6999 | 2.678 | 3.703 | 0.048 | 0.152 | 76.0 | 0.70698 | 0.00002 |
| 79.9 cpx3 | 3.1318 | 1.755 | 2.370 | 0.031 | 0.098 | 76.0 | 0.70686 | 0.00001 |
| 79.9 cpx4 | 5.2857 | 2.849 | 3.682 | 0.051 | 0.151 | 74.8 | 0.70712 | 0.00001 |
| 79.9 cpx5 | 1.7351 | 1.005 | 1.358 | 0.018 | 0.056 | 75.7 | 0.70712 | 0.00001 |
| 79.9 cpx6 | 3.7003 | 2.107 | 2.875 | 0.038 | 0.118 | 75.6 | 0.70713 | 0.00001 |
| 79.9 cpx7 | 2.9277 | 1.625 | 2.229 | 0.029 | 0.092 | 76.0 | 0.70716 | 0.00001 |
| 79.9 cpx8 | 7.5234 | 4.012 | 5.265 | 0.072 | 0.217 | 75.1 | 0.70712 | 0.00001 |
| CdT 108.3 | | | | | | | | |
| 108.3 cpx2 | 1.0561 | 0.643 | 0.948 | 0.012 | 0.039 | 76.5 | 0.70681 | 0.00001 |
| 108.3 cpx4 | 0.5970 | 0.308 | 0.431 | 0.006 | 0.018 | 75.0 | 0.70720 | 0.00002 |
| 108.3 cpx5 | 2.3090 | 1.390 | 2.287 | 0.025 | 0.094 | 79.0 | 0.70686 | 0.00001 |
| 108.3 cpx6 | 3.1160 | 1.914 | 3.292 | 0.034 | 0.135 | 79.9 | 0.70722 | 0.00004 |
| 108.3 cpx7 | 1.1160 | 0.634 | 0.980 | 0.011 | 0.040 | 78.4 | 0.70700 | 0.00001 |
| 108.3 cpx9 | 0.6920 | 1.179 | 1.944 | 0.021 | 0.080 | 79.2 | 0.70687 | 0.00002 |

* Mg# = 100 × molar Mg/(Mg + Fe); ** normalized to the recommended value of NIST SRM 987 (0.71025); *** 2 × standard error.

## 4. Results

### 4.1. Mineral Chemistry of CdT Pyroxene

Clinopyroxene from the three CdT units occurs as euhedral to anhedral dark to pale green phenocrysts (Figure 3). Some pyroxene crystals from the investigated Somma-Vesuvius lavas are slightly optically zoned from core to rim (Figure 4), and very few are reversely zoned. However, because the crystals were collected from crushed centimeter-sized rock chips, several analyzed individuals were fragments for which it was impossible to reconstruct the shape and the original position within the crystal. In sample CdT 49.5, most of the crystals have a pale green core surrounded by a dark green rim, indicating a reverse zoning (Figure 3A). In samples CdT 79.9 and CdT 108.3, normal zoning (dark green core surrounded by light green rim) is common, but also some crystals with complex zoning (alternating bands with variable colored tone from core to rim) occur (Figure 3B,C). In all the investigated samples, there are also normally zoned clinopyroxene crystals exhibiting cores characterized by a sieve texture (Figure 4).

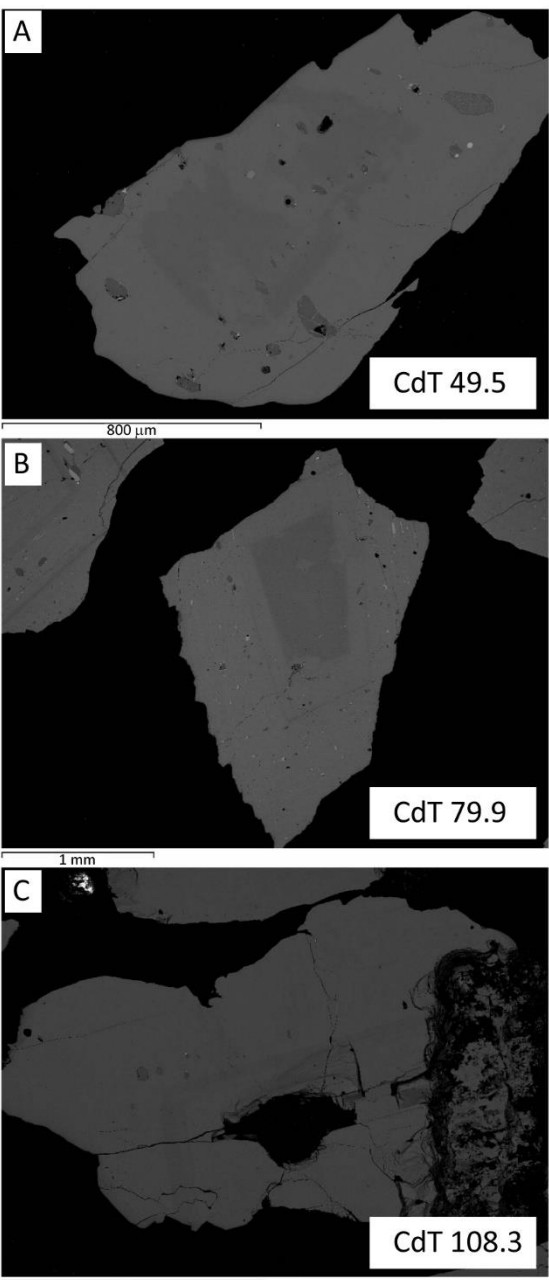

**Figure 4.** Back-scattered electron images of selected clinopyroxene phenocrysts from the investigated CdT units. (**A**) clinopyroxene from sample Cdt 49.5; (**B**) clinopyroxene from sample Cdt 79.9; (**C**) clinopyroxene from sample Cdt 108.3.

Based on the results of SEM-EDS analyses (Table S1 of Supplementary Materials), all analyzed crystals from the selected CdT samples are Ca-Mg-Fe ("Quad") clinopyroxene, having Q (=Ca + Mg + Mn + $Fe^{2+}$ atoms per formula unit (apfu)) in the range 1.88–1.99 and J (=2 × Na apfu) in the range 0.02–0.09, according to IMA/CNMMN rules [36]). They are classified as ferroan diopside ($Wo_{50-44}En_{49-35}Fs_{16-4}$) with the following adjectival modifiers: Aluminian, Ferroan, Ferrian (common); Aluminian, Ferroan (less common); Aluminian, Ferrian (rare); Chromian (very rare). The clinopyroxene crystals from sample CdT 108.3 have a composition variable from $Wo_{48}En_{46}Fs_6$ to $Wo_{46}En_{39}Fs_{14}$. Those from sample CdT 79.9 have a less variable composition, $Wo_{49-45}En_{47-37}Fs_{15-6}$. The composition of the clinopyroxene from sample CdT 49.5 is in the range $Wo_{50-44}En_{49-35}Fs_{16-4}$. In the quadrilateral classification diagram, the CdT clinopyroxene crystals fall within the compositional field of the Somma-Vesuvius pyroxenes from the literature (Figure 5).

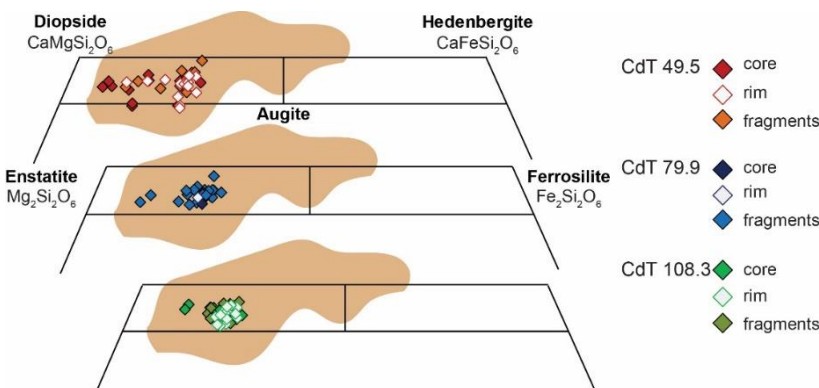

**Figure 5.** Portion of the quadrilateral classification diagram for pyroxenes ([36] and references therein) showing the compositional range of clinopyroxene from the three selected CdT samples. Data from Table S1 of Supplementary Materials. The light brown field encompasses the compositional range of Somma-Vesuvius clinopyroxene from the literature [5,37].

The whole range of Mg# (=100 × molar $Mg/(Mg + Fe^{2+} + Mn)$) is 92–69, with different ranges for the three analyzed samples: Mg# of clinopyroxene from sample CdT 108.3 ranges from 88 to 74; sample CdT 79.9 has clinopyroxene with Mg# variable between 89 and 71; clinopyroxene crystals from sample CdT 49.5 show the widest Mg# range, 92–69 (Table S1 of Supplementary Materials; Figure 6).

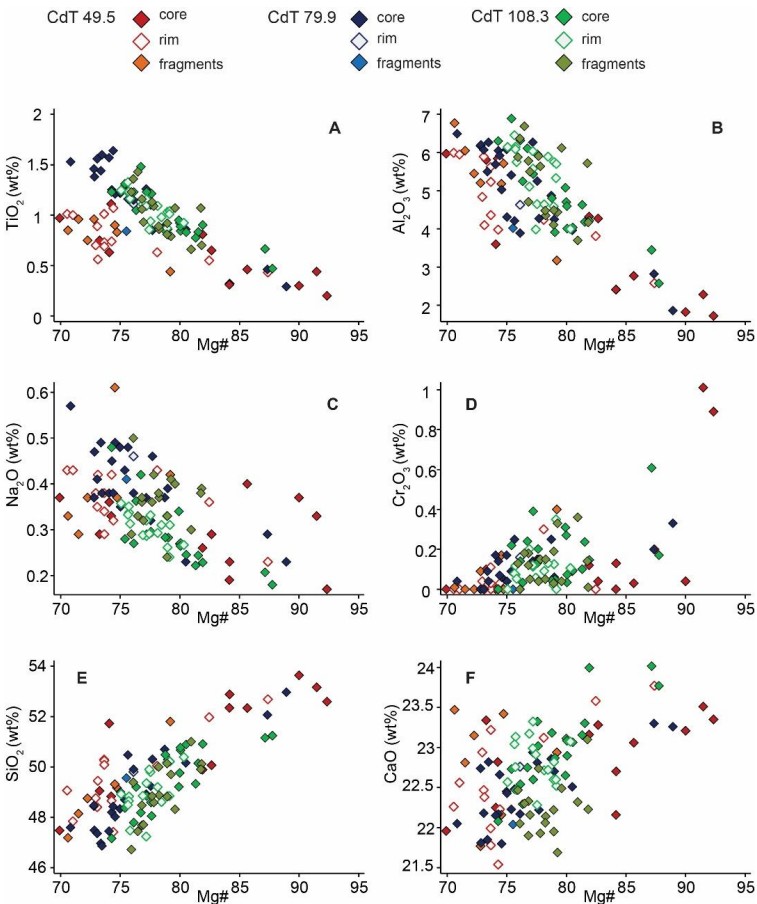

**Figure 6.** (**A**) Mg# vs. $TiO_2$, (**B**) Mg# vs. $Al_2O_3$, (**C**) Mg# vs. $Na_2O$, (**D**) Mg# vs. $Cr_2O_3$, (**E**) Mg# vs. $SiO_2$ and (**F**) Mg# vs. CaO of clinopyroxene crystals from the selected CdT samples. Data from Table S1 of Supplementary Materials.

TiO$_2$ and Al$_2$O$_3$ contents range from 1.64 to 0.20 wt% and from 6.89 to 1.72 wt%, respectively (Figure 6A,B). Na$_2$O and Cr$_2$O$_3$ contents range from 0.61 to 0.17 wt% and from 1.01 to 0 wt%, respectively (Figure 6C,D). The SiO$_2$ content ranges from 53.64 to 46.72 wt% and the CaO content ranges from 24.01 to 21.54 wt% (Figure 6E,F). TiO$_2$, Al$_2$O$_3$ and Na$_2$O contents increase, whereas Cr$_2$O$_3$ and SiO$_2$ contents decrease, with decreasing Mg#. CaO is quite constant.

### 4.2. Sr-Isotopic Composition of CdT Single Clinopyroxene Crystals

The $^{87}$Sr/$^{86}$Sr ratios of the single clinopyroxene crystals from the K-trachybasalt/shoshonite CdT 108.3 range from 0.70681 to 0.70722; those of clinopyroxene from the shoshonite CdT 79.9 range from 0.70686 to 0.70716, whereas the $^{87}$Sr/$^{86}$Sr of clinopyroxene crystals from the latite CdT 49.5 show higher values ranging from 0.70752 to 0.70781 (Table 1). Overall, the analyzed single clinopyroxene phenocrysts show Sr-isotopic disequilibrium with respect to their whole rocks (analyzed by Di Renzo et al. [16]), as will be illustrated in Section 5. Figure 7 shows that all the analyzed single clinopyroxene crystals have Mg# within the range detected through SEM-EDS analysis for each selected CdT sample.

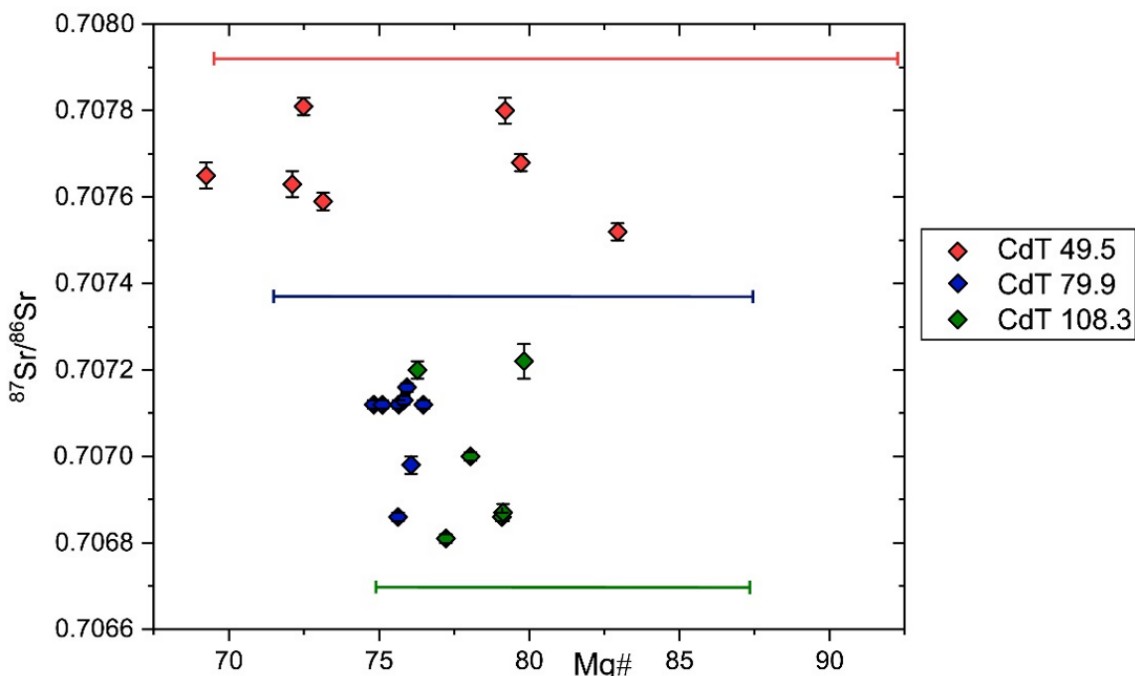

**Figure 7.** $^{87}$Sr/$^{86}$Sr ratio vs. Mg# of single clinopyroxene crystals from the selected CdT samples (data from Table 1). The thick colored horizontal bars represent the Mg# ranges of the clinopyroxene crystals determined through SEM-EDS analysis (Table S1 of Supplementary Materials).

## 5. Discussion

The Sr-isotopic composition of the Somma-Vesuvius products belonging to different periods of activity ranges between ~0.7062 and ~0.7090 [6,8,10–12,14,16–18,38–41]. Although the range of values is continuous, the distribution of the $^{87}$Sr/$^{86}$Sr ratios shows some peaks characterized by a high frequency of certain populations of values (Figure 8A).

Such a polymodal distribution indicates that many distinct magmatic components fed the magmatic system during the Somma-Vesuvius eruptive history. The Sr-isotope composition of products from different eruptions follows a systematic trend through the stratigraphic sequence (Figure 8A) that has been attributed to the arrival of isotopically different magma batches generated in a variable mantle source (e.g., [6,8,15,39,42,44,45]).

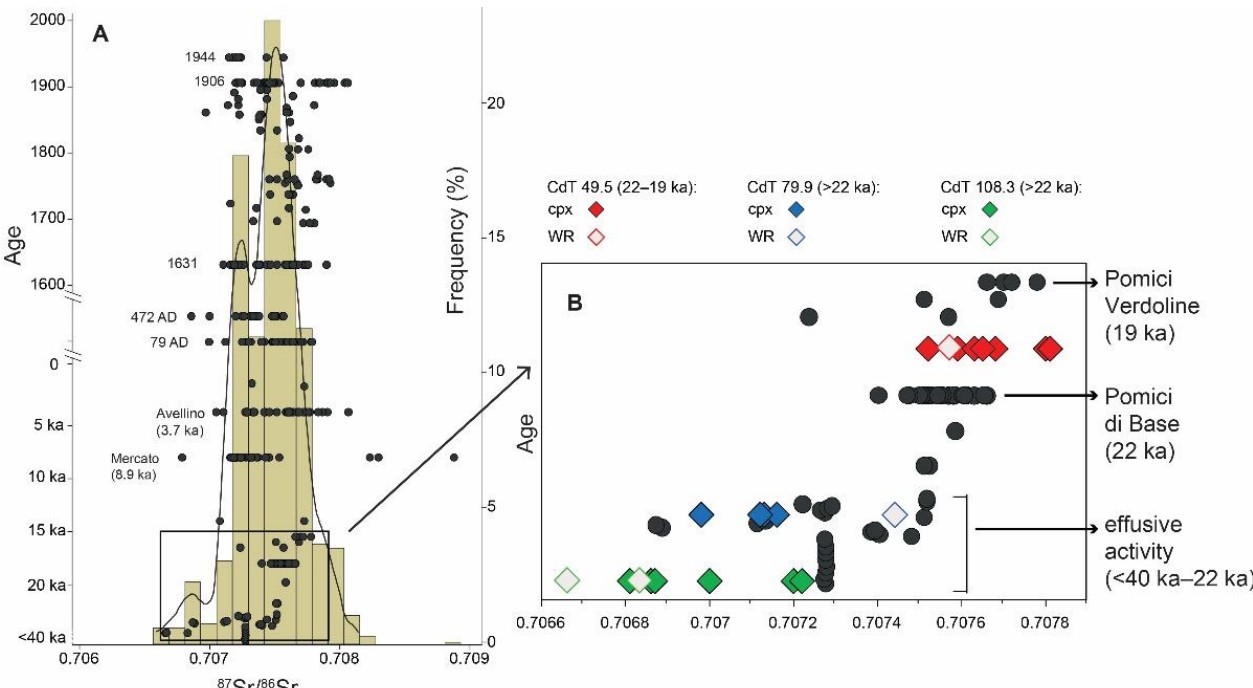

**Figure 8.** (**A**) $^{87}$Sr/$^{86}$Sr frequency histogram of Somma-Vesuvius products belonging to different periods of activity; (**B**) enlargement of the diagram to the left for the <40–19 ka time period, showing $^{87}$Sr/$^{86}$Sr ratio of clinopyroxenes from the selected samples and their whole rocks versus the stratigraphic position. Data from this paper and [6,8,10–12,14,16–18,38–43].

In particular, Figure 8B shows the predominance of a less radiogenic magmatic component ($^{87}$Sr/$^{86}$Sr < 0.7074) during the construction of the strato-volcano, which occurred, through the emplacement of lava flows and scoria deposits, between ca. 40 ka and 22 ka. The Pomici di Base Plinian eruption (22 ka) marked the shift towards a more explosive activity [2,20]. The event was preceded by lava flows that recorded an increase in the $^{87}$Sr/$^{86}$Sr ratio toward a more radiogenic magmatic component ($^{87}$Sr/$^{86}$Sr = 0.7077; Figure 8B). Such a trend, hence, is compatible with the arrival in the feeding system of a new magma with an isotopic signature different from the previous one and/or with assimilations of crustal rocks at mid-crustal depth where magmas stagnate. During the period of Mt. Somma's multistage caldera formation (22 ka–79 AD), in which mostly Plinian and sub-Plinian events occurred, the volcano's eruptions were fed by both less and more radiogenic magmatic components (Figure 8A). The Sr-isotopic heterogeneity of the products that erupted during this period well reflects the occurrence of magmatic processes able to change the isotopic signature of the primitive magmas. The isotopic variations result from both source heterogeneity and crustal assimilation at mid-crustal depth (ca. 8–12 km), where the Somma-Vesuvius magmas may have interacted with Hercynian continental crust and/or Mesozoic limestone [12,15,38,41,43,45–48]. At shallower depths, isotopically distinct magmas further differentiate and mix with each other [12,15,19,38,39,49]. The occurrence of such processes is supported by a great deal of data, e.g., compositionally zoned phenocrysts, distinct types of clinopyroxene coexisting in the same rocks, geochemical and isotopic variations within juvenile clasts from a single eruption and mineral/host rock disequilibrium. The chemical and isotopic mineral–melt disequilibria have been detected in the studied lavas as well as in the products belonging to the subsequent activity (e.g., [38,50]). In addition, the complex zoning detected in some cores (Figure 4) of clinopyroxene from all the analyzed samples is further evidence of disequilibrium, indicating that the core could have been affected by dissolution processes. Figure 8B shows that the single clinopyroxene crystals from samples CdT 108.3 and CdT 79.9 are in isotopic disequilibrium with their host rocks (data from Di Renzo et al. [16]). In particular, crystals from sample CdT 108.3 show Sr-isotopic ratios

higher than that of the whole rock, whereas crystals from sample CdT 79.9 show Sr-isotopic ratios lower than that of the whole rock. This suggests that crystals from sample CdT 108.3 grew in a magma with a more radiogenic Sr-isotopic ratio than that of the magma represented by the host rock. Similarly, clinopyroxenes from sample CdT 79.9 grew in a magma with a Sr-isotopic ratio lower than that of the magma represented by the host rock. In addition, crystals from sample CdT 49.5 partly show an isotopic disequilibrium with their host rocks, with crystals having values both considerably higher and slightly lower than those of the whole rocks.

In addition to the detected mineral–melt isotopic disequilibrium, most of the clinopyroxene crystals show chemical disequilibrium with the host rock composition as well. Aiming at testing the equilibrium between these minerals and the coexisting melt(s), we compared the normative mineral components, predicted for a mineral phase crystallized from a melt with composition similar to the bulk rocks, with those measured in the analyzed crystals (e.g., [51,52]). The measured Di (diopside) and Hd (Hedenbergite) components were calculated following the scheme proposed by Putirka [51]. The predicted clinopyroxene components based on melt composition were calculated using Equation (3.1a) by Putirka [53]. In a measured vs. predicted DiHd component plot, the equilibrium field is located inside the 1:1 $\pm$ 0.05 line (Figure 9).

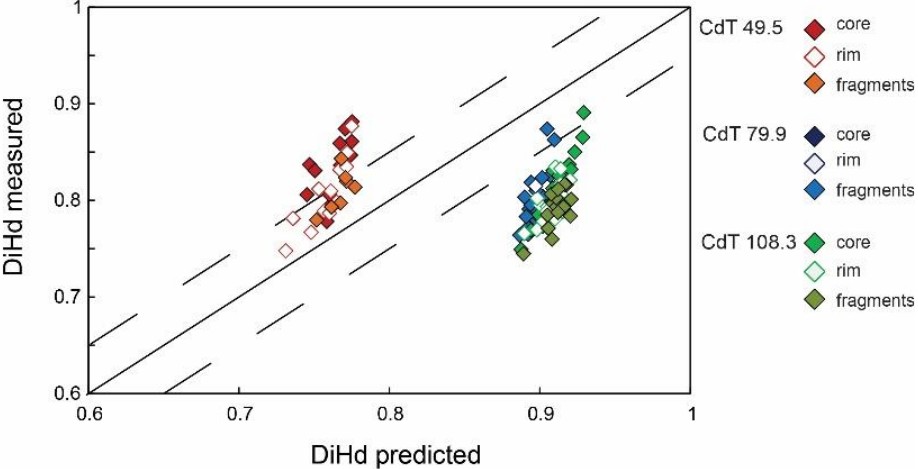

**Figure 9.** Comparison between amount of the measured DiHd clinopyroxene component and that predicted from the melt composition [52], for the selected CdT units.

The diagram illustrates that almost all crystals from samples CdT 108.3, CdT 79.9 and CdT 49.5 are outside the equilibrium field. This implies that the crystals were segregated from magma(s) with a different chemical composition with respect to that represented by the host rocks. The change in the composition of the magma should be simply due to a fractional crystallization process. Nevertheless, crystallization alone does not cause a change in the Sr-isotopic composition. In light of this, in addition to the chemical disequilibrium, the detected isotopic disequilibrium between the crystals and their host rocks (Figure 8) indicates that clinopyroxene crystals were growing during open-system processes able to change the chemical and isotopic composition of the crystallizing mineral phases. Alternatively, the analyzed clinopyroxene crystals may have been inherited from magmas with chemical and isotopic compositions different from those of their whole rocks.

Since the clinopyroxene retains the isotopic composition of the magma(s) from which it was segregated, the wide ranges of Sr-isotopic composition of the single crystals within each sample well reflect the occurrence of open-system processes, suggesting that the individual crystals crystallized from (i) isotopically distinct magmas, e.g., due to magma mingling/mixing processes, or (ii) from a magma that was changing its Sr-isotopic composition during crystal segregation, e.g., due to assimilation processes. The whole rock Sr-isotopic ratios also well reflect the shift in the Sr-isotopic composition with time. By taking into account the products that erupted between <40 and 22 ka (Figure 8B), the CdT

108.3 whole rocks record a less radiogenic composition ($^{87}$Sr/$^{86}$Sr = 0.7068), whereas the CdT 79.9 whole rock records one of the most radiogenic compositions ($^{87}$Sr/$^{86}$Sr = 0.7074) within that time period. This finding indicates the predominance of the less radiogenic magmatic component at the beginning and the prevalence of a more radiogenic component during the latest phases of the effusive activity preceding the Pomici di Base eruption. In addition, that feature suggests (i) input of magma with a more radiogenic composition or (ii) assimilation of material with high $^{87}$Sr/$^{86}$Sr. Since the samples CdT 79.9 and CdT 108.3 have clinopyroxene with similar $^{87}$Sr/$^{86}$Sr isotopic ratios, such processes could be related to a unique magmatic chamber that equilibrates with time (starting with CdT 108.3 whole rock composition and ending with CdT 79.9 whole rock composition). The $^{87}$Sr/$^{86}$Sr ratios detected in the clinopyroxene crystals of the analyzed lava units well show a trend of enrichment toward more radiogenic values with time, hence suggesting a progressive refilling of isotopically distinct magma batches in the Somma-Vesuvius plumbing system. At the end of the effusive activity preceding the Pomici di Base eruption, the erupted products also show a third Sr-isotopic component ($^{87}$Sr/$^{86}$Sr > 0.7077) with higher values with respect to those of the previous period. The most radiogenic component detected in clinopyroxene from sample CdT 49.5 and in the products of the Pomici di Base and Pomici Verdoline eruptions can be related to highest degrees of assimilation of crustal or wall rock during magma ascent or during stationing in the plumbing system, which causes an increase in the $^{87}$Sr/$^{86}$Sr ratios. An alternative, or even additional, reason for this increase should be the new inputs of magma that are more differentiated (as shown by the composition of both whole rock and clinopyroxene of sample CdT 49.5, with respect to those of samples CdT 79.9 and CdT 108.3, e.g., Figure 6) and have a more radiogenic Sr signature compared to magmas that erupted during previous periods (Figure 8), thus possibly derived from an enriched mantle source. Notably, the Sr-isotopic composition of the Pomici di Base, the sample CdT 49.5 and Pomici Verdoline also slightly decreases: this can also be caused by the input of a mafic magma component with a less radiogenic signature, like the one feeding the previous activity. Moreover, the input of new magma and processes of assimilation can also be responsible for gas enrichment (e.g., [43]), thus favoring a transition to explosive eruptive dynamics. Interestingly, similar mineralogical and isotopic disequilibria regarding clinopyroxenes and other mineral phases are common in alkaline potassic rocks from Ischia and Campi Flegrei (e.g., [54,55]), suggesting similar magmatic processes.

Sr-isotope microanalysis applied to some volcanic systems of the world highlighted the close connection between magma recharge, increasing volume of magma in shallow reservoirs and increasing explosivity of the eruptions. For example, at Stromboli volcano (south Italy), explosive eruptions occur periodically, about one to three events per year that are more energetic than the usual persistent activity of the volcano. The kind of explosive eruptions has been related to the arrival in the plumbing system of a volatile-rich mafic magma with low phenocryst content (the so-called LP-magma) that mixes with a high porphyritic magma with a distinct $^{87}$Sr/$^{86}$Sr ratio [31,56]. Varying degrees of complexity have been also identified by determining, through Sr isotope microanalysis, the chemical and isotopic composition of pumice-fall units from the Minoan cycle at Santorini, Greece [57]. The authors recognized relatively simple cases where crystals were in isotopic equilibrium with the host glass and much more complex cases where clear evidence of interaction between at least two liquid components and three different crystal populations were found, for instance in the case of the large caldera-forming Minoan eruption. A similar connection between magma recharge and explosivity of eruptions has been also suggested for several eruptions that occurred in the Neapolitan area (e.g., [19,20,22,23,43,58–62]). The above-mentioned literature studies and this work show that detailed information on the magmatic processes can be retrieved from an in-situ Sr-isotope investigation in single crystals. Such in-depth information bears important implications for a better knowledge of the past behavior of the volcano that must be taken into account for mitigating the volcanic risk in highly populated areas, such as that around the Somma-Vesuvius volcanic complex.

## 6. Conclusions

New compositional and Sr-isotopic data on separated clinopyroxene phenocrysts from selected units of the Camaldoli della Torre (CdT) cored sequence allowed us to shed light on the behavior of the Somma-Vesuvius magmatic feeding system in the course of its eruptive history. The CdT units investigated here are younger than 40 ka and formed during a period of effusive activity that led to the Mt. Somma strato-volcano growth, through the emplacement of K-trachybasalt/shoshonite to latite lava flows and scoria deposits. After this period, the Pomici di Base Plinian eruption at 22 ka marked a shift in the eruptive dynamics toward a more explosive character fed by generally differentiated magmas. The change in volcanic activity toward high-magnitude eruptions was preceded by a shift in the Sr-isotopic composition in products that erupted before the Pomici di Base eruption. In particular, an increase in the $^{87}Sr/^{86}Sr$ ratios has been detected through time in the clinopyroxene of the CdT lava units: this is explainable by the arrival of magma(s) with a distinct, more radiogenic isotopic signature in the Somma-Vesuvius magmatic system. The input of new magma(s) is also responsible for the chemical and isotopic mineral–melt disequilibria detected in the analyzed rocks.

Previous geochemical and isotopic studies have shown the role of magma mixing processes that occurred before and/or during the well-known Plinian and sub-Plinian eruptions at Somma-Vesuvius. Nevertheless, a systematic and detailed investigation of the isotopic (Sr-Nd-Pb) variations and mineral–melt disequilibria is lacking for the inter-Plinian periods of activity that preceded the major explosive events. These aspects are fundamental for comprehensive knowledge that can be integrated with that provided in previous geochemical studies to build a unique behavioral model of this both famous and dangerous volcano.

**Supplementary Materials:** The following supporting information can be downloaded at: https://www.mdpi.com/article/10.3390/min12091114/s1, Table S1: Chemical composition of CdT pyroxene crystals from SEM-EDS microanalysis; Table S2: Sr-isotopic ratios measured on the NIST-SRM 987 international standard with approximate Sr content of 3, 6 and 12 ng.

**Author Contributions:** Conceptualization, V.D.R. and L.C.; data curation, C.P., I.A., P.P. and M.D.; formal analysis, V.D.R., C.P. and P.P.; funding acquisition, L.C. and M.D.; investigation, V.D.R., C.P., I.A., L.C., P.P. and M.D.; methodology, V.D.R. and M.D.; supervision, M.D.; validation, V.D.R., L.C. and P.P.; visualization, V.D.R., C.P. and M.D.; writing—original draft, V.D.R., C.P., P.P. and M.D.; writing—review and editing, I.A. All authors have read and agreed to the published version of the manuscript.

**Funding:** This work was financially supported by EVG1-2001-00046 ERUPT European project. The research partially benefited from funding provided by the Italian Presidenza del Consiglio dei Ministri-Dipartimento della Protezione Civile (DPC; DPC-INGV V2 project). Scientific papers funded by the DPC do not represent its official opinions and policies. The INGV and OV laboratories have been also financially supported by the EPOS Research Infrastructure through the contribution of the Italian Ministry of University and Research (MUR).

**Institutional Review Board Statement:** Not applicable.

**Informed Consent Statement:** Not applicable.

**Data Availability Statement:** The data presented in this study are partly available in Supplementary Materials; all other data are available on request from the corresponding authors.

**Acknowledgments:** Roberto de Gennaro is thanked for assistance during SEM-EDS analytical sessions, and Lorenzo Fedele is thanked for advice during photomicrographs acquisition at the optical microscope at DiSTAR-Dipartimento di Scienze della Terra, dell'Ambiente e delle Risorse, Università di Napoli Federico II (Italy). We thank D. Morgan for helping in single crystal preparation and ICP-AES analysis at Dept. of Earth Science, Durham University.

**Conflicts of Interest:** The authors declare no conflict of interest.

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
