# Peer review of "Geochemical and Sr-Isotopic Study of Clinopyroxenes from Somma-Vesuvius Lavas: Inferences for Magmatic Processes and Eruptive Behavior"

_minerals, doi:10.3390/min12091114_

Round 1

Reviewer 1 Report

Review Summary

The submitted manuscript by Renzo et al. aims to constrain the behavior of the Somma- Vesuvius volcano and its magmatic feeding system along with the change in the eruptive dynamics. This work is certainly a welcome addition to the worldwide database for volcanic plumbing systems. It is nice that the authors have kept their discussion relatively brief, with the main conclusions largely supported by their data in combination with extant literature. I really enjoyed the flow and clarity of this manuscript. The authors have done a very good job in collecting a variety of generally good quality data from a difficult complex volcanic system. In conclusion, I believe the work presented here is important and worth publishing in a reputed journal like Minerals. The paper is well written with clarity but I still have some minor comments. I hope the authors find my comments useful in improving the presentation of their work.

The detailed comments (line by line) are mentioned below:  

Lines 30-56 - The introduction reads very well but I would suggest elaborating a bit more in order to provide the wholesome background to the problem addressed in this work.

Lines 57-82 - The petrological background of the different units in this section is not detailed well. To give readers a detailed idea of the petrological background, I will suggest the authors provide more insights with suitable references from studies carried out earlier.

Lines 89-98 - It appears to me that this whole section is written on the basis of the previous studies without citations. Authors should properly show the original source of this information in order to use these statements.

Lines 106-110 - What do the authors mean by bulk clinopyroxene crystals and single crystals…please elaborate.

Lines 102-110 - This whole section should go to the introduction. It does not seem to fit here.

Line 161 - A complementary analysis of the Nd isotopes along with the Sr isotopes would have served better to distinguish the various open and closed system processes and even have allowed for assessing the characteristics of the source regions of the magmas generated eventually leading the authors to identify which processes played a critical role in the causing the present chemical composition of the magma and their isotopic values.

Line 177 - Describe the optical characteristics observed under the microscope such as grain size, shape, and the optical zoning pattern displayed in plane polarised and cross polarised light.

Line 178 - A-line profile from the core to rim will clearly depict the kind of compositional variation and thus the processes such as magma mixing with the same composition or different composition.

Line 178-180 - This line does not make sense to me I can’t follow what the authors actually want to say here.

Line 181 SEM-EDS data are semiqualitative and semiquantitative and involve large uncertainty. Thus, EPMA analysis of the grain separates will provide a more robust mineral chemical variation which is highly required to asses any open or closed system processes occurring within any volcanic system, as minerals archive the complex processes occurring right from their formation to emplacement.

Line 216 although the authors have analyzed single clinopyroxene grains for Sr isotopic studies but it still masks the actual values. It is noted that there are normal and reverse zoning within the single grains thus there could be isotopic variations among the core and rim portions of the grain. Hence an in-situ isotopic analysis of the pyroxene grains will distinctly resolve whether the mixing was between different magmas derived from different sources or between various batches of similar magma or crustal assimilation.

Line 256 - but could also be due to assimilations of crustal rocks while pooling.

Line 298-302- It is one of the prominent assessments in the discussion part. How do authors correlate chemical and isotopic disequilibrium? Also, how do these characters suggest the growth of clinopyroxene during open system processes? Can authors precisely talk about the process governing such heterogeneity in chemical and isotopic signatures?

Line 300- 302 Such processes are common in the volcanoes but which magma mixed with which one has not been specified in the discussion. A clear picture of the various complex processes is required for better interpretation.

Line 315-316 Grammatical error… Please correct as ‘The kind of explosive eruptions has been related…’

Line 344 – 345 the more radiogenic values of 87Sr/86Sr could also be due to the assimilation of the crustal rocks in shallow crustal chambers before the eruption.

Line 347 – 348 whether the new magma batch was of a similar composition to the previous one or a distinctly new one with a different composition. The chemical and isotopic mineral-melt disequilibria observed due to different magmas can be explained in little more detail in the discussion part.

The manuscript uses an integration of good datasets to address the crucial research problem taken up and the study deserves to be published in high-rank journals. After incorporating these minor comments, the manuscript can be recommended for publication.

Author Response

In the attached file our ponit-by-point response to both Reviewers 'comments are reported.

Thanks a lot for all the suggestions and observations that help us to get better our paper. 

Best regards, 

all the authors 

Reviewer 2 Report

The manuscript is well written, and has some new analysis of previous analyzed samples of a previous published paper (Di Renzo et al., 2007; J. Petrol., 48:753-784).

Considering the knowledgment of the studied samples, a better discussion is considered for this manuscript. Several comments are indicated in the attached PDF file. However the main suggestions are:

1.- In the abstract, there is no indication about the main findings of the work, nor the range of 87Sr/86Sr ratios by unit that are discussed in the manuscript.

2.- BSE images can be used to establish the zoning that are indicated in the text.

3.- A better conceptual model is needed in the discussion section. This is the most remarkable comments in the PDF file. The discussion is poor and misses the evidence of the establishment of a mafic recharge (indicated using the Francalanci et al's papers). What is observed, in Figure 8b, is that the oldest stages (Cdt 79.9 and 108.3) have lower 87Sr/86Sr values, which increases towards the youngest units (CdT 49.5, Pomici di Base and Pomici Verdoline). Furthermore, both Pomici di Base and Pomici Verdoline have a decreasing of Sr-isotope ratios related to CdT 49.5. Thus, a mafic recharge can be related to both Pomici's, with the effusive CdT 49.5 lavas considered as the contaminant agent, whilst equilibrium is reached from the oldest (CdT 108.3) to the middle lavas (CdT 79.9) suggesting a steady stage during evolution of the Mt. Somma lava flows.

Author Response

We are grateful to the reviewer for the helpful comments which allowed to improve the quality of our work. 

We better addressed the discussion with the implication of our data. Please see replies to the following comments and to comments of reviewer 1.

1.- In the abstract, there is no indication about the main findings of the work, nor the range of 87Sr/86Sr ratios by unit that are discussed in the manuscript.

Comment on the Abstract in the annotated pdf: There's nothing about the behavior of the magmatic system. The abstracts tells nothing about the findings of the paper. Which processes did occur? What is the observed change? Also, what compositional Sr-isotopes differences (in ranges) were obtained?

In agreement with the reviewer, we think that the abstract should report something about the detected compositional and isotopic ranges and about the main findings of the paper, but without “spoiler” the final results and conclusions.  We added some of such information in the abstract: 

“Somma-Vesuvius is one of the most dangerous among the active Italian volcanoes, due to the explosive character of its activity and because it is surrounded by an intensely urbanized area. For mitigating the volcanic risks, it is important to define how the Somma-Vesuvius magmatic system worked during the past activity and what processes took place. Knowing the behavior of the  magmatic system is indeed crucial for both interpretating any change in the dynamics of the volcano and predicting its future behavior. A continuous coring borehole drilled at Camaldoli della Torre, along the southern slopes of Somma-Vesuvius, allowed reconstructing its volcanic and magmatic history in a previous study. In this work, new  the wide range of chemical compositional (Mg#=92–69) and Sr isotopic data (87Sr/86Sr=0.70781–0.70681) composition, have been collected on single clinopyroxenes separated from selected lava flow units of the Camaldoli della Torre sequence, characterized by different ages and bulk geochemical composition. The new data, have been integrated with the already available bulk geochemical and Sr isotopic data,. The detected chemical and isotopic signature and their variation through the time allowing us to better constrain the behavior of its magmatic feeding system, highlighting thate mixing and/or assimilation processes occurred before a significant change of the eruptive dynamics at Somma-Vesuvius during a period of polycyclic caldera formation, started with the Pomici di Base Plinian eruption (22 ka).

2.- BSE images can be used to establish the zoning that are indicated in the text.

In the new version of the manuscript, we made use of the BSE images together with Figure 3 to better describe the zoning pattern of the clinopyroxene crystals from the different analyzed samples. Please see reply to comment about line 177 of reviewer 1.

3.- A better conceptual model is needed in the discussion section. This is the most remarkable comments in the PDF file. The discussion is poor and misses the evidence of the establishment of a mafic recharge (indicated using the Francalanci et al's papers). What is observed, in Figure 8b, is that the oldest stages (Cdt 79.9 and 108.3) have lower 87Sr/86Sr values, which increases towards the youngest units (CdT 49.5, Pomici di Base and Pomici Verdoline). Furthermore, both Pomici di Base and Pomici Verdoline have a decreasing of Sr-isotope ratios related to CdT 49.5. Thus, a mafic recharge can be related to both Pomici's, with the effusive CdT 49.5 lavas considered as the contaminant agent, whilst equilibrium is reached from the oldest (CdT 108.3) to the middle lavas (CdT 79.9) suggesting a steady stage during evolution of the Mt. Somma lava flows.

We better explained the implication of our data, by providing a better conceptual model of the various complex processes that can be responsible of the chemical and isotopic variations detected in the single clinopyroxenes from the analyzed lava units. Please see reply to comment of revewer1 about line 300– 302 and line 347–348.

Other comments retrieved from the annotated pdf by Reviewer 2

Comment (line 56): Figure 2 should be placed after this paragraph.

We thank the Reviewer for this suggestion. So, we moved Figure 2 to the end of the Introduction.

Comment: This figure (referred to Figure 2) is important in the description. Thus, you should use it more in the paragraph above.

Following this suggestion, we have called up this figure more times in the description of the various stratigraphic units in the revised manuscript (line 86 and lines 111-115).

Comment (line 97): Is this your work? If not, references must be indicated.

This part of the volcanological and petrological background synthetizes results of the thorough petrological characterization of the Camaldoli della Torre borehole products from Somma-Vesuvius made by Di Renzo et al., 2007. Actually, we put the citation [16] at the beginning and at the end of the paragraph, but that was not enough, probably. In the revised manuscript, the text has been changed to recall many times the source of the statements (lines 118-128).

Comment (line 127): It is better to use (or indicate) the measure in the SI system (m, mm, cm, etc.).

In the annotated pdf of the manuscript the symbol @ appears, whereas the symbol f (Greek letter phi) was put in our original manuscript. Actually, f is currently used as a unit for the size of sieves employed for granulometric analysis. Anyway, we used the measure in the SI system in the new version of the manuscript. We changed the related sentence in:

“The selected lava samples were gently crushed to lapilli-size grains through a jaw crusher. Samples were sieved using a stack of sieves with meshes ranging from – 1f to 1f 0.5 to 4 mm.”

Comment (line 132): What is the average size (of the crystals)? Standard deviation?.

We added this information at lines 155-157:

“The average size of the crystals from sample CdT 108.3 is 1.6 ±0.3 mm, that of clinopyroxenes from sample CdT 79.9 is 2.9 ±0.6 mm, whereas that of crystals from sample CdT 49.5 is 1.3 ±0.3 mm. “

Comment (caption of Fig.4): What does this show? What is it useful? Maybe to indicate zoning of another mineral characteristics.

In the previous version of the manuscript, Figure 4 was used only to show that crystals are zoned, whereas in the new version, the zoning pattern of the crystals has been described in detail and this figure well show some kind of complex zoning in crystals from all the analyzed samples, that is also indicative of disequilibrium. We added such consideration in the text at lines 307-309:

 “The chemical and isotopic mineral-melt disequilibria have been detected in the studied lavas as well as in the products belonging to the subsequent activity (e.g., [38, 51]). Also the complex zoning detected in some cores (Figure 4) of the clinopyroxenes from all the analyzed samples is a further evidence of disequilibrium, indicating that the core could have been affected by dissolution processes. Figure 8b shows that the single clinopyroxene crystals from samples CdT 108.3 and CdT 79.9 are in isotopic disequilibrium with their host rocks (data from Di Renzo et al. [16]).”

Comment on Figure 8b:

  • Sr-isotope ratios do not decrease within cpx crysts, how can it be related to mafic inputs?
  • Less radiogenic clinopyroxene that can be related to less degrees of contamination and/or new mafic inputs. As both flows (CdT 79.9 and 108.3) have cpx with similar 87Sr/86Sr isotopic ratios, this could be related to a unique magmatic chamber that equilibrates with time (starting with CdT 108.3 whole rock composition and ending with CdT 79.9 whole rock composition).

In the new version of the manuscript, we better explained that the increase of the Sr-isotope ratios could be related to input of magma with a more radiogenic signature, possibly derived from an enriched source and/or to crustal assimilation processes. We also used these useful comments to better explain our model (please see lines 350-387).

Comment (line 298): From figure 9??? Mostly all pyroxene lie outside the equilibrium field, from all samples. This is also constrained from Figure 8b, which tells more than you indicated.

We modified the text at lines 336-338:

“The diagram illustrates that only few almost all crystals from samples CdT 108.3, andCdT 79.9 and CdT 49.5 are in outside the equilibrium field, whereas almost half of the clinopyroxenes from sample CdT 49.5 is inside the equilibrium field.”

In the new version of the manuscript, we also added further consideration about the disequilibrium features (e.g., line 338-349).

Comment (line 318): Francalanci et al. indicated antecryst with 87Sr/86Sr values lower than yours at CdT49.9, but similar to the lowest values of CdT 79.9 and 108.3. Thus, to determine if mafic recharge exists for Somma-Vesuvius, the tepha units should be analyzed. Maybe, what you have is the most evolved end-member that interacts with the mafic magmas (most differentiated lavas).

According to this observation, in the new version of the manuscript, based on the increasing of the Sr-isotopic composition through the time, we explained our model in detail (e.g. at lines 371-387): contrary to the Francalanci et al. model, we proposed that the system was fed by a magma that is more evolved and more radiogenic with respect to those that fed the activity preceding the Pomici di Base eruption. Please see also reply to next comments.

Comment (line 327): Indeed. Mafic recharge is considered as triggering of volcanic eruptions, and changes from effusiveness to explosivity of magmas. However, Figure 8b indicates that younger clinopyroxene crystals are more radiogenic and not less radiogenic, as expected if they are related to mafic recharge. What is your explanation for this?

Actually, a mafic magma can also show a more radiogenic composition with respect to an evolved magma, as commonly occurs in the products of the nearby Campi Flegrei caldera. Anyway, in the new version of the manuscript, we better explained that the magma component feeding the activity immediately preceding 22 ka is more evolved and more radiogenic in comparison to that feeding the previous period. Moreover, starting from 22 ka, the slight decrease of the Sr-isotopic composition can also be explained with the input of less radiogenic mafic magma.    

Comment (line 332): This discussion needs further development and more support.

What is observed in figure 8b can be related to Figure 6. As average, clinopyroxenes of CdT 49.5 have more SiO2 and less TiO2 and Al2O3 than the others. Thus, this pyroxenes were generated in a more differentiated magmas.

Furthermore, the whole rock Sr-isotope ratio composition of CdT is more or less the average of these clinopyroxenes. Thus, these cpx are evidencing the processes occurring in this magmatic chamber.

We are grateful for this suggestion. In fact, in the new version of the manuscript, at lines 373-382, we better explained that:

“The most radiogenic component detected in the clinopyroxenes from sample 49.5 and in the products of the Pomici di Base and Pomici Verdoline can be related to higher degrees of assimilation of crust or wall rock during magma ascent or during stationing in the plumbing system, that causes an increase of the 87Sr/86Sr and/or to new inputs of magma that is more differentiated (as shown by the composition of both whole rock and clinopyroxene of sample CdT 49.5, with respect to those of samples CdT 79.9 and CdT 108.3, e.g., Fig. 6) and with a more radiogenic Sr signature compared to magmas erupted during previous periods (Fig. 8), thus possibly derived from an enriched mantle source. […]”

Comment (lines 341-344): Then, what does cause the explosive character of this more radiogenic magmas is the assimilation of felsic material rather than mafic inputs? This contradicts what is exposed above, as you put Strimboli as example.

In the new version of the manuscript, we better explained that both assimilation and mixing can be responsible of the detected chemical and isotopic variations and that such processes are able to cause explosive character, e.g., at lines 383-387:

“[…] this can also be caused by input of the mafic magma component with a less radiogenic signature, like that feeding the previous activity. Moreover, the input of new magma and processes of assimilation can also be responsible of gas enrichment (e.g., [43]), thus favoring a transition to explosive character of the eruptive dynamics.”

The new version of the manuscript is in the attached file.

Best regards,

the authors

Round 2

Reviewer 2 Report

Dear Authors

The manuscript shows an improvement. I appreciate that you had considered all the comments and suggestions made.

Regards